## TECHNIQUE

# VasoTracker 2: Open-source software and hardware for tracking blood vessel diameter and assessing vascular function

Matthew D. Lee[1] [ID], Christopher Osborne[2,3] [ID], Ross Stevenson[1], Amy MacDonald[1], Grace Ebner[4] [ID], Danielle A. Jeffrey[5] [ID], Margaret A. MacDonald[1], Xun Zhang[1] [ID], Charlotte Buckley[1,6] [ID], Fabrice Dabertrand[5,7] [ID], Daniel R Machin[8], Jason Au[9] [ID], Osama F. Harraz[4] [ID], Nathan Tykocki[10] [ID], John G. McCarron[1] [ID] and Calum Wilson[1] [ID]

[1]*Strathclyde Institute of Pharmacy and Biomedical Sciences, University of Strathclyde, Glasgow, UK*
[2]*Faculty of Science, University of Strathclyde, Strathclyde, UK*
[3]*SUPA School of Physics & Astronomy, University of Glasgow, Glasgow, UK*
[4]*Department of Pharmacology, Larner College of Medicine, Vermont Center for Cardiovascular and Brain Health, University of Vermont, Burlington, VT, USA*
[5]*Department of Anesthesiology, University of Colorado Anschutz Medical Campus, Aurora, CO, USA*
[6]*Centro de Biología Integrativa, Universidad Mayor, Santiago, Chile*
[7]*Department of Pharmacology, University of Colorado Anschutz Medical Campus, Aurora, CO, USA*
[8]*Department of Cell Biology and Physiology, University of New Mexico School of Medicine, Albuquerque, NM, USA*
[9]*Department of Kinesiology and Health Sciences, University of Waterloo, Waterloo, ON, Canada*
[10]*Department of Pharmacology and Toxicology, Michigan State University, East Lansing, MI, USA*

Handling Editors: Kim Barrett & T. Alexander Quinn

The peer review history is available in the Supporting Information section of this article (https://doi.org/10.1113/JP289322#support-information-section).

This article was first published as a preprint. Lee MD, Osborne C, Stevenson R, MacDonald A, Ebner G, Jeffrey DA, Macdonald MA, Zhang X, Buckley C, Dabertrand F, Machin DR, Au J, Harraz OF, Tykocki N, McCarron JG, Wilson C. 2025. VasoTracker 2: An open-source platform for quantitative analysis of vascular reactivity and function. bioRxiv. https://doi.org/10.1101/2025.04.23.648411

**Abstract figure legend** VasoTracker 2 is an open-source blood vessel diameter tracking software and hardware for assessing vascular function. VasoTracker 2 combines custom software with specialized hardware to track blood vessel diameter changes in real time. The software works with bright-field, fluorescence and ultrasound imaging to measure vessels ranging in diameter from 10 μm to 1 mm. The hardware includes a pressure myograph vessel chamber, pressure controller and monitoring equipment. Researchers can integrate the software with existing equipment, or can build a complete pressure myography system to study how vessels respond to pressure, flow, drugs and more. Both the software and hardware designs are open-source with complete build and installation instructions provided, making blood vessel research tools more accessible and affordable.

**Abstract**   VasoTracker 2 is a collection of open-source tools for blood vessel diameter measurement and vascular physiology research, featuring automated diameter tracking software and low-cost myography hardware components. This blood vessel analysis platform addresses the need for accessible alternatives to expensive commercial vascular imaging systems by providing comprehensive tools for studying vessel reactivity and endothelial function. The software enables multipoint diameter tracking in branched vessels using advanced edge detection algorithms, supporting brightfield microscopy, fluorescence imaging and ultrasound recordings. Automated pressure protocols enable standardized myogenic tone experiments, and both real-time and offline analysis of pre-recorded data are supported. The platform's versatility allows researchers to study vascular dynamics across diverse experimental conditions, from isolated vessel preparations to *in vivo* imaging applications. For *ex vivo* applications, VasoTracker 2 includes modular, low-cost open-source pressure myograph hardware: a confocal-compatible vessel chamber and a programmable pressure controller, VasoMoto. These components enable construction of a complete pressure myograph system at a significantly reduced cost compared to commercial alternatives. The hardware integrates seamlessly with the diameter measurement software. VasoTracker 2 provides free software and low-cost hardware for vessel diameter analysis, broadening access to advanced vascular research tools. The platform's open-source nature allows researchers to modify and extend the system for specific applications. This flexibility, combined with significant cost savings, benefits vascular researchers worldwide.

(Received 23 May 2025; accepted after revision 16 September 2025; first published online 17 October 2025)

**Corresponding authors** M. Lee and C. Wilson: Strathclyde Institute of Pharmacy and Biomedical Sciences, University of Strathclyde, 161 Cathedral Street, Glasgow, G4 0RE, UK.     Email: matthew.lee@strath.ac.uk, c.wilson@strath.ac.uk

### Key points

- VasoTracker 2 is an open-source platform that combines versatile vessel diameter tracking software with modular low-cost hardware components for vascular physiology research.
- The software enables multipoint diameter analysis across brightfield, fluorescence and ultrasound imaging modalities.
- The hardware includes a confocal-compatible vessel chamber and programmable pressure controller (VasoMoto) that can be assembled into a complete pressure myograph system.
- The open-source approach provides researchers with accessible tools for advanced vascular physiology experiments.

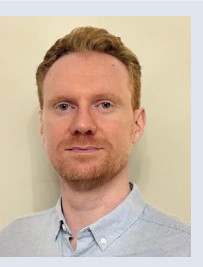

**Matthew Lee** is a Chancellor's Fellow at the University of Strathclyde, where he investigates vascular endothelial communication in cardiovascular disease. His current research focuses on understanding how endothelial cell networks sense and respond to various simultaneously arriving stimuli. His research combines advanced calcium imaging with network analysis methods.

## Introduction

The regulation of blood vessel diameter is a key determinant of blood flow control, vascular resistance and tissue perfusion. Disruptions in these control mechanisms are implicated in almost all human diseases, either as a cause, a consequence or as an aggravating factor. For example, reductions in vessel diameter contribute to hypertension and are linked to cardiovascular disease (Wilson et al., 2019); plaque deposits impair cerebral autoregulation in Alzheimer's disease patients, reducing blood flow and worsening disease progression (Taylor et al., 2024); and excessive vasodilatation leads to shock in sepsis patients (Jarczak et al., 2021). Understanding these dysfunctional blood vessel dynamics is essential for uncovering disease mechanisms and developing effective treatments.

Quantitative analysis of vascular dynamics requires sophisticated tools that can accurately measure blood vessel diameter across diverse experimental contexts. Although multiple methodologies exist for studying vascular function, ranging from isolated vessel preparations to *in vivo* imaging, all share a fundamental need for precise, reliable diameter measurement. Despite the critical importance of these measurements in vascular research, accessible tools for quantitative analysis across different experimental modalities have been limited. Researchers often rely on expensive commercial systems with restrictive proprietary software, or custom-developed solutions using macros in ImageJ (NIH, Bethesda, MD, USA) (Fernández et al., 2014) or scripts in MATLAB (MathWorks Inc., Natick, MA, USA) (O'Herron et al., 2016, 2022). This fragmented landscape raises significant barriers to entry for new researchers.

To overcome these barriers, we previously released VasoTracker, an open-source platform initially focused on making pressure myography - the gold standard technique for studying blood vessel regulation (Schjørring et al., 2015; Schubert et al., 2023; Wenceslau et al., 2021) - more accessible. The initial proof-of-concept analysis software was paired with a low-cost, open-source pressure myograph that could match the performance of significantly more expensive commercial systems in assessing blood vessel diameter (Lawton et al., 2019). Although our initial focus was on pressure myography, the research community's response revealed an unexpected insight: the software's value extended far beyond our original vision. Despite its preliminary nature, VasoTracker software was adopted by vascular research laboratories worldwide and has been used to study many aspects of blood vessel physiology. For example, VasoTracker has been used to examine vasoconstriction and endothelium-dependent vasodilatation (Moreira et al., 2023), the physiology of vasospam (Ng et al., 2024), and vascular dysfunction in obesity and in cardiovascular disease (Edwards et al., 2021;

Zheng, Li et al., 2023; Zhang et al., 2025). The software has been used to study isolated blood vessel function in arteries from mice (Cheon et al., 2021; Zheng, Berg Sen et al., 2023; Zheng et al., 2025), rats (Wilson et al., 2020) and pigs (Morton et al., 2022). VasoTracker software has also been used to assess blood vessel function *in vivo*, in zebrafish (Hoffmann et al., 2022) and in humans (Seredyński et al., 2021). Moving beyond vascular studies, VasoTracker software has been used to investigate contractions in the mouse renal pelvis (Grainger et al., 2022) and bladder biomechanics (Hennig et al., 2023).

The widespread adoption and creative application of VasoTracker beyond its initial scope has driven demand for advanced features to support more diverse experimental designs. Users have requested advanced features including standalone analysis software for multiple imaging modalities, improved algorithms for complex vessel geometries, compatibility with various camera systems and automated experimental protocols. These community needs prompted the development of VasoTracker 2 - a complete reengineering of both software and hardware components. At its core, VasoTracker 2 features versatile software for blood vessel diameter tracking across multiple imaging modalities, complemented by modular hardware that, although designed in line with our original vision of pressure myography, can also be used in other systems.

Here, we introduce the new capabilities of VasoTracker 2 that address these community needs. We demonstrate how its software enables quantitative analysis across multiple imaging modalities, including brightfield microscopy, fluorescence imaging and ultrasound, and how it supports advanced applications such as multipoint diameter assessment in branched vessels and arterial strips. We also illustrate how the new modular hardware components facilitate specialized applications such as automated pressure myography experiments and confocal calcium imaging in blood vessels. These advances collectively transform VasoTracker from its initial implementation into a versatile platform for investigating complex vascular dynamics in both traditional and novel experimental contexts.

## Methods

### Ethical approval

All animal and human experiments described in this study were conducted with appropriate institutional ethical approval and in accordance with relevant national guidelines, the *Declaration of Helsinki* and the ARRIVE guidelines 2.0 (Percie du Sert et al., 2020), as detailed in the Experimental Protocols sections below. Specifically, animal work was approved by the University of Strathclyde Animal and Welfare Ethical Review Committee (UK

Home Office regulations), University of Colorado Anschutz Medical Campus IACUC, University of Vermont IACUC and University of Utah/Salt Lake City Veterans Affairs Medical Centre Animal Care and Use Committee. Human work was approved by the University of Waterloo ethics board (ORE 22 477).

## VasoTracker 2 system overview and availability

VasoTracker 2 is a modular open-source system designed to measure blood vessel diameter and assess vascular function. The system comprises versatile diameter-tracking software that works with multiple imaging modalities (brightfield, fluorescence and ultrasound) and complementary low-cost hardware components for pressure myography experiments. The software enables real-time vessel tracking, automated pressure protocols, and post-experiment analysis of pre-recorded images. For researchers working with isolated vessels, the modular hardware components can be combined to create a complete pressure myography setup or integrated individually with existing laboratory equipment.

The software, hardware design files, component lists, assembly instructions, user manuals and tutorial videos are all freely available through the VasoTracker website (https://vasotracker.com). Users can source components from standard suppliers and using the designs on the VasoTracker website, either machine parts themselves at their Departments mechanical workshop or use commercial machining services. This modular approach allows researchers to adopt selected elements based on their specific experimental needs and existing equipment.

## VasoTracker 2 software

The VasoTracker 2 software is an advanced tool for tracking the diameter of blood vessels in pressure myography experiments and beyond. Distributed as a standalone executable, requiring no installation or proprietary software, with comprehensive user documentation and tutorial videos available on the Vaso-Tracker repositories (https://vasotracker.com/software), the software offers enhanced capabilities for tracking diameter under a range of experimental procedures (brightfield and fluorescence imaging) and for recording user interventions.

VasoTracker 2 software supports five primary functions:

- **Image acquisition and recording**: Captures and records images of pressurized blood vessels using a wide range of microscope-attached digital cameras.
- **Real-time diameter analysis**: Calculates, graphs and records blood vessel diameter in real time.
- **Pressure control and data acquisition**: Controls pressure-servo systems via an analogue-digital converter for automatic pressure control and collects data from Arduino-based temperature and pressure sensors.
- **User input logging**: Allows users to log interventions, such as the addition of biological compounds to the vessel chamber.
- **Pre-recorded image analysis**: Facilitates diameter measurement in pre-recorded images, enabling flexible post-experiment analysis.

The graphical user interface (GUI) is split into four parts: a control panel for adjusting key camera and analysis settings; an interactive live graph panel displaying auto-scrolling diameter measurements; a real-time image feed with diameter indicators; and a data entry table for logging experimental treatments. The interface allows users to configure settings, view dynamic data and record experimental notes during live experiments. Additional settings are accessible through the File menu. Further menus for controlling advanced settings (e.g. controlling the graph display, inputting the pressure protocol) are accessible through the program's menu bar.

The software is implemented in Python 3 and built on Python bindings of the C++ core of the μManager microscopy control system (Edelstein et al., 2014; Stuurman et al., 2010). The software employs a modular Model-View-Controller architecture linking the μManager bindings with VasoTracker's 2 tracking algorithms and the GUI and facilitating easy extension (e.g. support for new cameras). Key software packaged include: NumPy (Harris et al., 2020), scikit-image (Walt et al., 2014), Matplotlib (Hunter, 2007), Christopher Gohlke's Tifffile (Gohlke, 2024) and Marcos Duarte's Detect Peaks function (Duarte & Watanabe, 2021). The complete source code is freely available, enabling users to modify and extend the software for specialized applications.

## Camera compatibility and settings

VasoTracker 2 software supports a wide range of digital cameras. Default support is provided for Basler (Basler AG, Ahrensburg, Germany) and Thorlabs (Thorlabs Inc., Newton, NJ, USA) cameras, which can be readily configured using pre-defined settings. Additionally, VasoTracker 2 provides native support for μManager configuration files, allowing the integration of a wide variety of additional cameras. Essential exposure control is handled by VasoTracker, but other camera settings (e.g. bit depth, binning, gain) may be configured in μManager and exported to VasoTracker via the configuration file. By default, VasoTracker limits data acquisition to 5 Hz, but

this may be increased to ~15 Hz on suitably equipped computers.

### Calibration

VasoTracker 2 requires calibration to convert pixel measurements to physical units. Users calibrate the software using reference objects of known dimensions (e.g. calibration slides) to establish the pixel-to-micron scaling factor for their specific imaging setup. This calibration approach works across brightfield microscopy, fluorescence imaging and ultrasound systems. Once calibrated, users can save these scaling factors for different imaging configurations and quickly switch between them as needed.

### Tracking algorithms and regions of interest

The original VasoTracker software measured vessel diameter in brightfield images using horizontal scanlines. VasoTracker 2 software maintains this fundamental approach (i.e. detecting vessel edges by identifying peaks in the derivative of intensity profiles) but builds upon the original algorithm by adding the ability to limit vessel tracking to five user-defined regions of interest or custom-drawn lines. These can be oriented horizontally, vertically, or the user can track diameter in up to five custom lines drawn at any angle. This derivative-based edge detection method provides precise localization of intensity transitions at vessel boundaries, enabling accurate diameter measurements even for small vessels (see section below on 'Choice of imaging system'). For fluorescent samples, VasoTracker 2 introduces a new algorithm tailored to their unique intensity profiles. Such enhancements may allow users to examine multiple vessel branches simultaneously or avoid artefacts like vessel side-branches or residual adherent tissue.

### Data export

VasoTracker software exports data in comma-separated value (CSV) format, ensuring compatibility with all common data analysis packages. Each experiment generates two CSV files: one containing diameter and sensor measurements, and the other with information that was entered into the table (e.g. user notes on drug additions, automatically controlled pressure changes, etc.). Users can choose to record images, both raw and with tracking indicators overlaid; these images are saved in Tag Image File Format (TIFF). Additionally, users can add experimental notes in the built-in VasoTracker 2 notepad, which are saved as a Text File Document (TXT).

### VasoTracker 2 hardware

To complement the software's versatile analysis capabilities, VasoTracker 2 includes modular hardware components for researchers working with isolated blood vessels: a pressure myograph vessel chamber, a low-cost, 3D printed peristaltic pump (VasoMoto) and a temperature and pressure monitor. These hardware elements can be used together to create a complete pressure myography system or integrated individually with existing laboratory equipment. Users wishing to add VasoTracker pressure myograph capabilities to an existing microscope may do so for a cost of ~£3000 at the time of writing (assuming external machining), with substantial cost reductions possible for institutions with in-house machining capabilities or bulk orders.

Full assembly instructions, parts lists, technical specifications and implementation considerations are available online (https://vasotracker.com/hardware).

### VasoTracker 2 pressure myograph chamber

The pressure myograph chamber design represents a significant departure from our original. The updated version is a modular bath consisting of a base and a low-volume (~2 mL) vessel chamber insert. The low volume chamber is optimized for efficient use of reagents and precise environmental control via super-fusion. The base can be machined from POM-C (i.e. acetal copolymer), and the chamber insert from acrylic, via any commercial CNC (i.e. computer numerical control) machining service. These materials ensure durability and compatibility with various experimental conditions. The modular base is designed to fit a Thorlabs MLS series motorized scanning stage but can be directly placed on any manual microscope stage or easily adapted for other automated stages.

The bath is equipped with dual MPH3 pipette holders (WPI Inc., Sarasota, FL, USA), mounted to miniaturized three-axis translation stages (DT12XYZl; Thorlabs Inc.) via rotatable probe clamps (MCU; Siskiyou, Grants Pass, OR, USA). This configuration facilitates independent XYZ positioning of both cannulas with precise micro-metric control (~6 µm resolution), offering enhanced precision and flexibility. This setup makes it easier to align and manipulate blood vessels during mounting. It also allows blood vessels to be positioned near the bottom of the chamber, adjacent to the glass cover slip, for low-working distance or oil objective use on inverted microscopes. Additionally, the angled cannula design provides clearance for water dipping objectives on upright microscopes. Superfusion plumbing can be securely and flexibly attached using magnetic holders

## Pressure control

Three options are provided for pressure control. The first is VasoMoto, a newly developed manual pressure servo system. VasoMoto is a low-cost, open-source, 3D printed peristaltic pump specifically designed for pressure myography. VasoMoto consists of an Arduino-based microcontroller (Arduino, Monza, Italy), a custom instrumentation amplifier with a 16-bit analogue-to-digital converter and a precision stepper motor. The enclosure and pump head are 3D printed with minimal additional hardware. This unit interfaces directly with VasoTracker 2 to share continuous pressure measurement. VasoMoto is also capable of oscillating pressure to mimic the pulsatile pressure changes seen *in vivo* to rates as fast as the murine heart rate ($\sim$400 oscillations $min^{-1}$). Because Arduino-based micro-controllers can directly interface with a computer without the need for additional hardware, VasoTracker 2 can acquire pressure data without the need for an external interface or additional software.

VasoTracker 2 also supports automated pressure regulation using PS-200 Pressure Servo Controllers (Living Systems, St Albans, VT, USA), which are programmable via a National Instruments DAQ system (National Instruments, Austin, TX, USA) for precise control when intraluminal flow is not required. Finally, VasoTracker is compatible with traditional setups that use height-adjustable reservoirs connected to the glass cannulas in the myography bath. These reservoirs rely gravity to generate create a pressure gradient across the vessel, allowing luminal flow when required (Lawton et al., 2019).

## Temperature and pressure monitor

The VasoTracker 2 system combines temperature and pressure monitoring. The chamber's temperature is continuously monitored by a 10k negative temperature coefficient thermistor, with real-time readings displayed on an LCD and reported to the VasoTracker software. When pressure is managed using VasoMoto, pressure readings are transmitted directly to the VasoTracker software. When height adjustable reservoirs are used (for experiments with luminal flow), pressure levels may be monitored by flow-through pressure transducers, providing real-time pressure readings on an LCD screen and transmitting data to the VasoTracker software.

## Choice of imaging system

The myograph chamber is designed for versatility and is compatible with most microscope setups. Data presented here were acquired using various microscopes, including Eclipse Ts2R (Nikon, Tokyo, Japan) and CK40 (Olympus, Centre Valley, PA, USA) inverted microscopes equipped with 4$\times$ and 10$\times$ objectives and complementary metal-oxide-semiconductor (CMOS) cameras (CS165MU/M; Thorlabs Inc.; or ACE a2A1920-160umBAS; Basler AG) and an FN-1 upright microscope with a 10$\times$ water-dipping objective (Nikon). Both recommended CMOS cameras have 3.45 µm pixels. Fluorescence imaging data were acquired using a 20$\times$ objective with a 13 µm pixel EMCCD camera (iXON Ultra 888; Andor, Belfast, UK). These combinations provide:

- 4$\times$ objectives: $\sim$0.86 µm/pixel sampling, $\sim$2.8 µm optical resolution, field of view up to 1.66 $\times$ 1.04 mm
- 10$\times$ objectives: $\sim$0.345 µm/pixel sampling, $\sim$1.3-1.5 µm optical resolution, field of view up to 662 $\times$ 414 µm
- 20$\times$ objectives: $\sim$0.17 µm/pixel sampling, $\sim$0.75 µm optical resolution, field of view up to 331 $\times$ 207 µm
- 20$\times$ oil immersion objective with EMCCD: $\sim$0.65 µm/pixel sampling, $\sim$0.37 µm optical resolution, field of view up to 666 $\times$ 666 µm

Importantly, with modern 3.5 µm pixel width cameras, the optical resolution (diffraction limit, determined by the numerical aperture of the objective) rather than pixel sampling typically limits measurement precision. Regardless, diameter changes on the order of 1–2 µm may be reliably detected, representing the practical limit imposed by optical resolution rather than digital sampling.

## Experimental protocols

To assess performance, we tested the VasoTracker 2 software across a range of experimental conditions and imaging modalities. The software was evaluated in both real-time vessel tracking with the VasoTracker 2 pressure myograph and offline image analysis of datasets generated using other experimental techniques that include brightfield, fluorescence and ultrasound imaging.

## Animals

All animals were housed with *ad libitum* access to food and water. Arteries were obtained from male (8–12-week-old) Sprague–Dawley rats at Strathclyde. These animals were killed by cervical dislocation with secondary confirmation via decapitation in accordance with Schedule 1 of the Animals (Scientific Procedures) Act 1986. Some experiments included male mice in Michigan and female mice in Vermont (indicated in the respective figure legends). Mice were killed by I.P. injection of sodium pentobarbital (100 mg $kg^{-1}$) followed by immediate decapitation. Ultrasound experiments were performed on anesthetized rats at the University of Utah as previously described (Machin et al., 2016).

### Humans

All human data presented in this study are collected from healthy human participants who provided informed written consent prior to data collection. The acquisition protocols were approved by the University of Waterloo ethics board (ORE 22 477) and complied with the *Declaration of Helsinki* regarding the use of human participants, except for registration in a database.

### Pressurized artery myography

Following death, mesenteric arcades were removed and transferred to a physiological salt solution of the following composition (mм): 125.0 NaCl, 5.4 KCl, 0.4 $KH_2PO_4$, 0.3 $NaH_2PO_4$, 0.4 $MgSO_4$, 4.2 $NaHCO_3$, 10.0 HEPES, 10.0 glucose, 2.0 sodium pyruvate, 0.5 $MgCl_2$ and 1.8 $CaCl_2$ (adjusted to pH 7.4 with NaOH). Third and fourth order mesenteric artery segments were mounted in a VasoTracker 2 pressure myograph and visualized via light microscopy, at 60 mmHg (unless otherwise indicated) and 37°C. For confocal calcium imaging experiments, arteries were loaded with the calcium indicator, Cal-520/AM (5 μм) and visualized using an Aurox Clarity structured illumination LED confocal imaging system (Aurox Ltd, Abingdon, UK). In other experiments, arteries were loaded with Calcein (1 μм) and visualized via epifluorescence microscopy using LED illumination (PE-4000; CoolLED, Andover, UK). One set of experiments examined vascular reactivity in flat-mounted (*en face*) arteries. In these experiments, arteries were cut open and the endothelium was preferentially loaded with Cal-520/AM and visualized using wide-field epifluorescence microscopy. In experiments examining cortical capillary-parenchymal arterioles preparations, experiments were performed as described previously (Jeffrey et al., 2022). Data from the flat-mounted arterial preparation and the parenchymal arteriole preparations were analysed using the offline analysis feature of VasoTracker 2.

### Flow-mediated dilatation

**Animals.** Rats were anaesthetized under 3% isoflurane and 100% oxygen in a closed chamber anaesthesia machine for ∼1–3 min. Anaesthesia was maintained using a nose-cone, and rats were secured in a supine position on a heated examination table to maintain body temperature that was equipped with ECG (VisualSonics, Toronto, ON, Canada). Briefly, an occlusion cuff (Harvard Apparatus, Fairfield, NJ, USA) was placed proximal to the right ankle. The ultrasound Doppler probe was placed proximal to the occlusion cuff on the superficial femoral artery. Measurement of superficial femoral artery vessel diameter was performed using a high-resolution micro-ultrasound imaging system (VisualSonics) equipped with an ultra-high frequency linear array transducer operating at 30–70 MHz (VisualSonics). The sample volume was maximized according to vessel size and was centred within the vessel on the basis of real-time ultrasound visualization. The superficial femoral artery was insonated and measurements of diameter were recorded at rest. After which, the occlusion cuff was inflated to occlude the distal tissue for 5 min. Artery diameter was recorded continuously throughout the occlusion period and for 3 min after cuff release. End-diastolic, ECG R-wave-gated images were collected from the ultrasound machine for off-line analysis of superficial femoral artery vasodilatation using Vaso-Tracker 2.

**Humans.** The brachial artery of a 21-year old female was imaged in B-mode with an 9–12 MHz linear array ultrasound probe (Vivid iq; GE Healthcare; Chicago, IL, USA) in the longitudinal plane (3 cm depth, 32 Hz frame rate). A pneumatic cuff positioned on the distal forearm was inflated to an occlusion pressure of 200 mmHg (minimum of 50 mmHg above systolic blood pressure) for 5 min, and subsequently released while images were continuously collected for an additional 3 min during reactive hyperaemia and stored for subsequent offline analysis using VasoTracker 2.

## Results

VasoTracker 2 is a completely redesigned open-source, modular platform for quantitative analysis of vascular reactivity and function across multiple experimental contexts. At its core is versatile analytical software that can track blood vessel dynamics in real-time or from pre-recorded images, complemented by customizable hardware components for specialized applications. This integrated approach enables researchers to analyse vascular function in diverse experiments.

### VasoTracker 2 software

The VasoTracker 2 software (Fig. 1*A*) provides comprehensive analysis of blood vessel diameter and function across multiple imaging modalities. Built on Python 3 and μManager 2, the platform operates in both real-time and offline modes, enabling researchers to track vessel dynamics from brightfield microscopy, fluorescence and ultrasound images. This integration with any μManager-supported camera allows Vaso-Tracker 2 to work with existing microscope systems, making sophisticated vascular measurement techniques accessible for analysing vascular reactivity from isolated vessel preparations to *in vivo* recordings (Fig. 1*B*).

Key features of VasoTracker 2 software:

- **Multimodal vessel tracking**: Analyse vessel diameter in real-time or from pre-recorded images across brightfield, fluorescence and ultrasound imaging modalities (Supplementary Movie 1).
- **Customizable analysis approaches**: User-defined regions or scan lines for edge detection to determine blood vessel inner diameter, outer diameter and wall thickness with precision (Supplementary Movie 2).

- **Hardware integration capabilities**: Seamlessly integrates with VasoMoto or commercially available pressure controllers for automated experiments.
- **Universal camera compatibility**: Functions with any camera supported by µManager 2, eliminating hardware restrictions.

A comprehensive list of components, design files, software and instructions for building and operating the system are available from the VasoTracker website

## A  VasoTracker 2 user interface

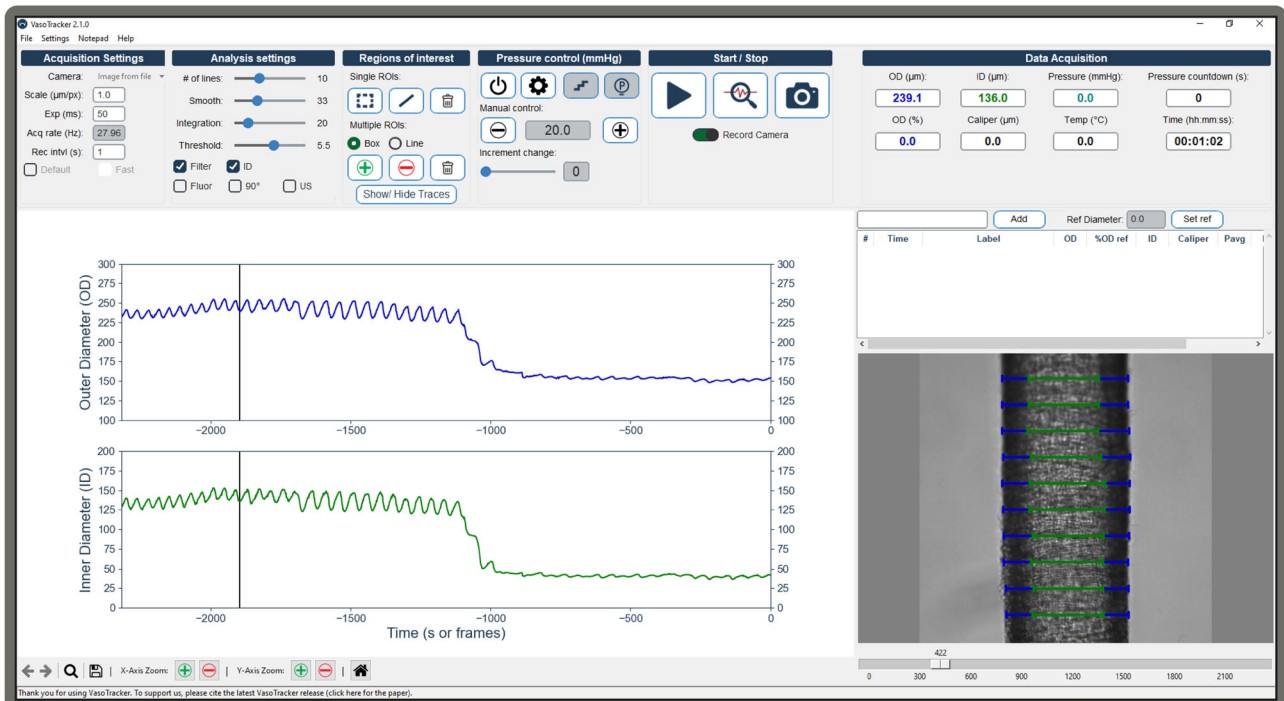

## B  Example applications

 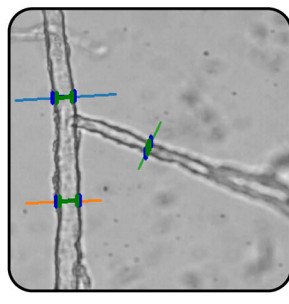 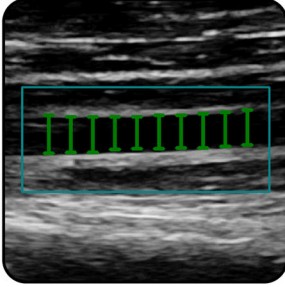 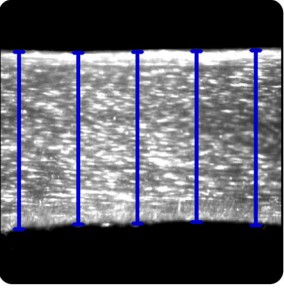

Pressurized artery (brightfield imaging) · Microvascular network (brightfield imaging) · Conduit artery (ultrasound imaging) · Arterial strip (fluorescence imaging)

**Figure 1. VasoTracker 2 software and tracking algorithms**
*A*, screenshot of the VasoTracker 2 graphical user interface. The software consists of four main panels: the settings pane, a live graph of outer and inner diameter, a table for recording experimental details and a live or pre-recorded view of the blood vessel with diameter indicators overlaid. *B*, example images showing VasoTracker diameter tracking in brightfield, fluorescence and ultrasound modes, highlighting the versatility across different vessel preparations and imaging applications. The blue box represents a user-selected region of interest.

(https://vasotracker.com) and GitHub repository (https://github.com/VasoTracker/VasoTracker-2).

## VasoTracker 2 software applications

The original VasoTracker software (Lawton et al., 2019) established an open-source foundation for recording and analyzing bright-field images of blood vessels obtained via pressure myography, enabling live assessment of vasoconstriction, endothelium-dependent vasorelaxation, propagated vasodilatation and myogenic responses. Vaso-Tracker 2 preserves these core functions while introducing new features that expand the range of vascular research possible with the open-source software. These include the ability to analyse pre-recorded video stacks (Fig. 2*A*), and new algorithms for multipoint/region of interest

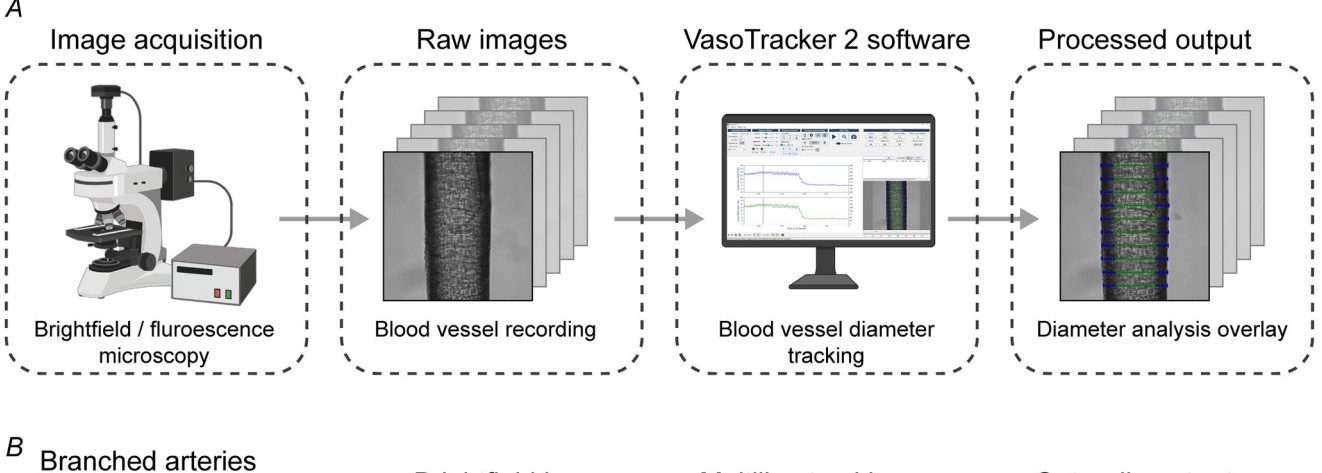

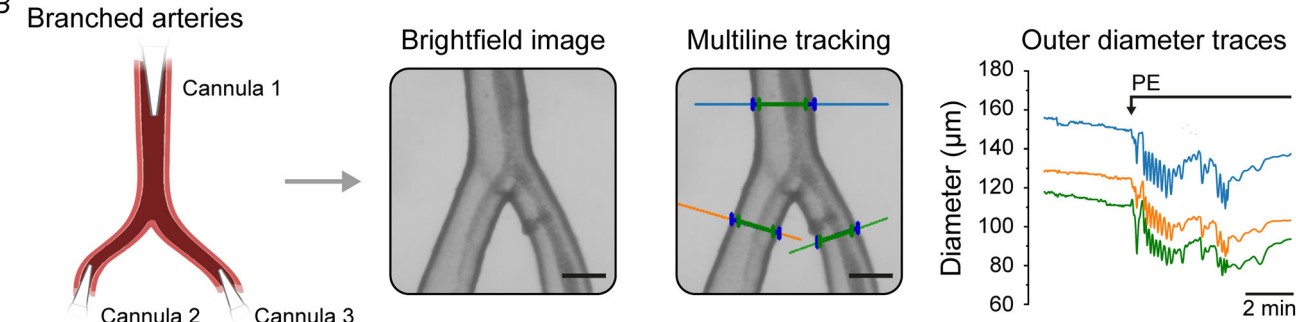

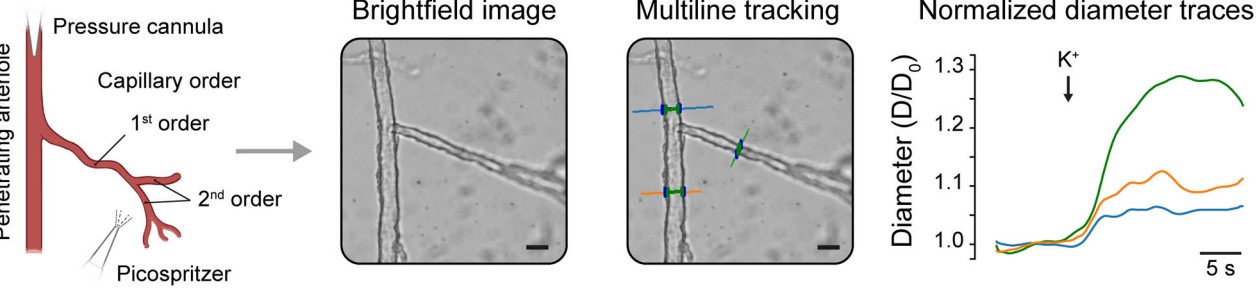

**Figure 2. Multipoint tracking of diameter in branched blood vessels**
*A*, chematic of the protocol for assessing blood vessel diameter in pre-recorded images. *B* and *C*, schematics, example images and diameter traces illustrating the multiline tracking feature in VasoTracker 2 in a triple-cannulated branched mesenteric artery (*B*) and in the capillary-parenchymal arteriole preparation (*C*). The data in (*B*) were obtained in a live experiment using fourth-fifth branch order mesenteric arteries of the rat. The third cannula was introduced to the bath and held in place using a magnetic attachment, and the artery was stimulated with phenylephrine (PE) (1 μM) to elicit vasoconstriction. Panel *C* shows the same algorithm applied to a pre-recording of capillary-parenchymal arteriole preparation from the mouse brain. In this experiment, 10 mM K$^+$ was picospritzed onto the capillary network of the parenchymal arterioles which initiated vasodilatation throughout the vascular network. Scale bars = 50 μm (*B*) or 20 μm (*C*). Components of this figure were created using bioRender.

diameter tracking in images obtained using multiple different imaging modalities. The following sections detail these new capabilities and their applications in addressing complex questions in vascular biology.

## Multipoint diameter assessment in branched blood vessels

The complexity of vascular networks requires simultaneous tracking of multiple vessel segments, particularly in branched structures such as cerebral parenchymal arterioles and corresponding capillary branches. The ability to track diameter changes at multiple points within these branched vessels is crucial for understanding how local and systemic factors influence the control of blood flow in the network. VasoTracker 2 introduces a multipoint tracking feature that allows concurrent assessment of diameter changes at multiple locations within a branched vessel and vascular network. This feature is shown in Fig. 2, where it was applied to two distinct experimental setups.

In the first (Fig. 2*B*), VasoTracker 2 software was used to measure the diameter of three arms of a triple-cannulated branched mesenteric artery (70 mmHg). The artery was stimulated with phenylephrine via bath perfusion, and the diameter of each branch was independently tracked during the live experiment. In the second setup (Fig. 2*C*), VasoTracker's multiline tracking algorithm was applied to pre-recorded images of mouse parenchymal arterioles with attached capillary network (the CaPa preparation) (Jeffrey et al., 2022). In this experiment, potassium (10 mM) was picospritzed onto the capillary network. VasoTracker was used to track the diameter changes that occurred in the capillary branch, and at two locations in the parenchymal arteriole: one upstream and one downstream of the branch point.

The ability to track diameter changes in multiple branches provides insights into the coordinated vascular responses that may occur in complex networks.

## Diameter tracking in fluorescence microscopy experiments

Fluorescence labelling is commonly used in vascular studies to enhance the visualization of blood vessels. For example, fluorescent dyes permit diameter tracking at the same time as visualizing the internal elastic lamina separating endothelial cells and smooth muscle cells (Garland et al., 2017) and to visualize blood vessels *in vivo* (Longden et al., 2017). To demonstrate the capabilities of VasoTracker 2's algorithms in tracking vessel diameter in fluorescence labelled blood vessels, we loaded pressurized mesenteric arteries with Calcein (1 μM) and used Vaso-Tracker 2 to assess vascular reactivity to phenylephrine

(1 μM) and acetylcholine (0.1–10 μM). Representative images and diameter traces are shown in Fig. 3*A*, demonstrating that the algorithm accurately tracks both the inner and outer walls of the artery.

Fluorescence labelling also facilitates the measurement of vascular reactivity in flat-mounted (*en face*) arterial strip preparations, a well-established method for assessing vascular reactivity (Huang et al., 2001; Wilson et al., 2020; Zhang et al., 2023, 2025). Example images and a trace of vascular reactivity from a flat-mount mesenteric artery are shown in Fig. 3*B*. In this experiment, Vaso-Tracker 2's algorithm was used to analyse pre-recorded images of flat-mount preparations recorded on an inverted fluorescence microscope. The images were loaded into the software, and the same algorithm was employed for fluorescence labelled pressurized vessels with the exception that 'inner diameter' measurement was not acquired. By incorporating these tracking algorithms into VasoTracker 2, we make fluorescence-based blood vessel tracking more accessible to the vascular research community.

## Diameter tracking in ultrasound imaging

Large conduit artery diameters are most commonly measured using diagnostic ultrasound because of its widespread availability, temporal resolution for dynamic vascular responses and non-invasive nature. Dynamic diameter tracking is particularly relevant for measures of arterial compliance (Engelen et al., 2015) and endothelial function, mostly commonly assessed via the flow-mediated dilatation technique as the relative change in diameter after a shear stress stimulus (Thijssen et al., 2019). Structural features of conduit arteries are also linked to future cardiovascular disease risk; for example, the thickness of the intima–media complex of the common carotid artery (Touboul et al., 2012).

VasoTracker 2's diameter tracking algorithms can be applied to ultrasound images, allowing for analysis of large conduit arteries in both animal models and clinical recordings (Fig. 3*C*). The software's ability to process pre-recorded images makes it suitable for retrospective analysis of ultrasound data, and allowing analysis of other datasets, providing researchers and clinicians with an open-source tool for quantitative assessment of vascular function in various cardiovascular conditions.

## Hardware integration

Beyond tracking vessel diameter in vascular imaging datasets, VasoTracker 2 software includes features for seamless integration with hardware components. The software can communicate with various pressure control systems, including commercial controllers, traditional

gravity-based setups and the new open-source options described below. This integration enables automated experimental protocols and continuous data logging from temperature and pressure sensors, allowing researchers to precisely control experimental conditions while simultaneously tracking vessel dynamics.

## VasoTracker 2 hardware

Building on these integration capabilities, and our continuing focus on open-source pressure myography, VasoTracker 2 includes modular hardware components for *ex vivo* applications. These components include a redesigned vessel chamber, temperature and pressure

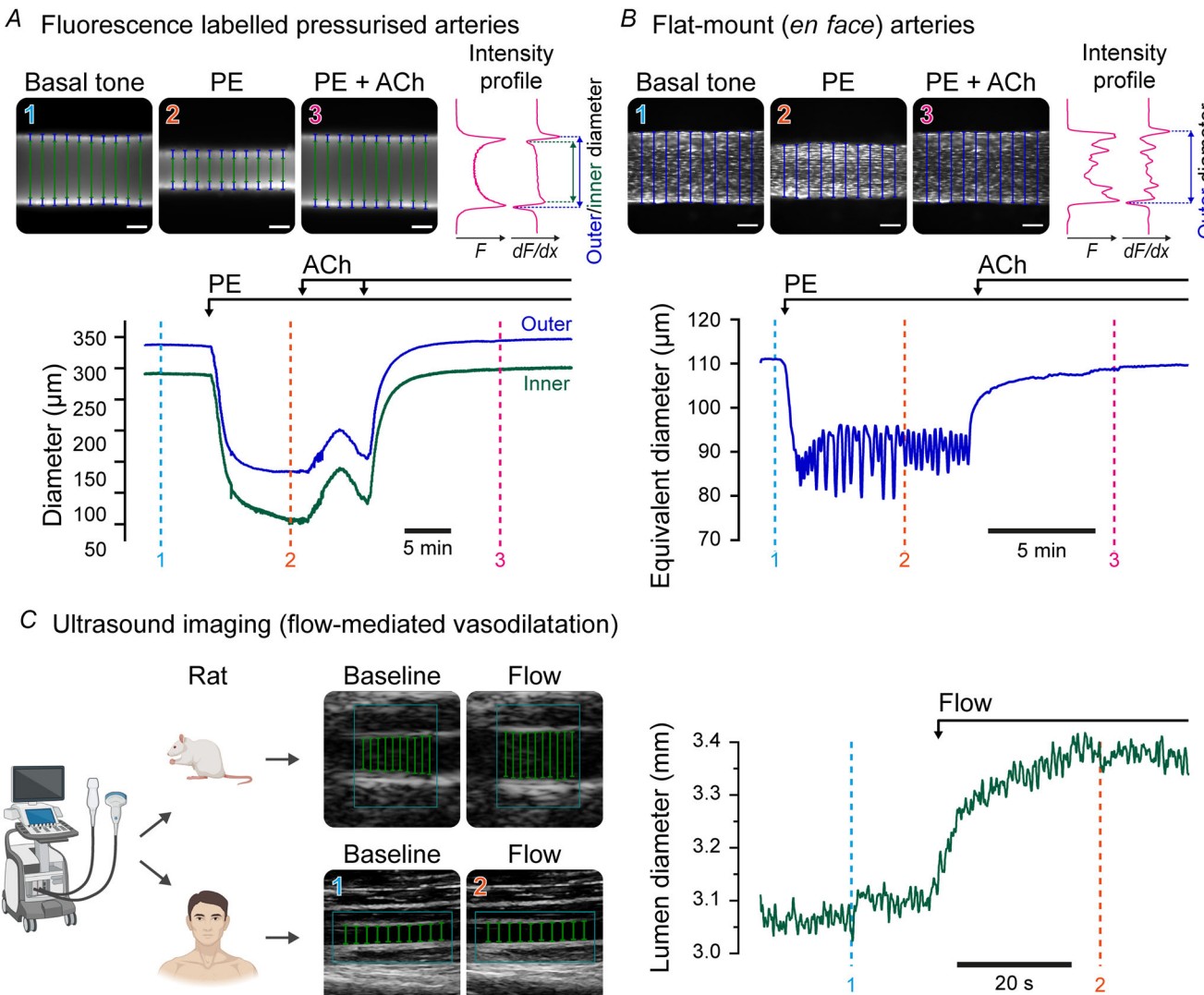

**Figure 3. Assessing vascular reactivity using fluorescence labelled vessels and ultrasound imaging techniques**
*A* and *B*, Still frame images and diameter traces of (*A*) fluorescence labelled pressurized arteries (Calcein, 1 μM) and (*B*) fluorescence labelled flat-mounted mesenteric arteries (Cal-520/AM, 1 μM). In each case, arteries were stimulated with phenylephrine (1 μM) to elicit vasoconstriction, and then acetylcholine [10 nM increased to 1 μM (*A*) or 1 μMM (*A*)] to induce endothelium-dependent vasodilatation. The positions of artery edges are determined from intensity profiles (*F*, upper panels) and their derivatives (dF/dx) using the new VasoTracker 2 algorithm. When assessing flat-mounted (*en face*) arteries, inner diameter detection is prevented and vascular reactivity is expressed as the equivalent diameter of a pressurized vessel. Scale bars = 100 μm. *C*, Still frame images showing flow-mediated vasodilatation in rat superficial femoral and human brachial arteries. In each case, ultrasound images were collected during flow-mediated dilatation experiments and stored for offline analysis using VasoTracker. Scale bars = 5 mm. *D*, A full trace of brachial artery diameter from a human participant from a flow-mediated dilatation test. Components of this figure were created using bioRender.

monitoring systems and a new programmable pressure controller, VasoMoto. These modular components can be combined to create a complete pressure myograph system at a fraction of the cost of commercial alternatives (Fig. 4*A* and *B*). Each component can also be used independently or integrated with existing laboratory equipment to enhance various vascular research applications.

Key features of VasoTracker 2 hardware:

- **VasoMoto pressure control**: Supports manual and programmable blood vessel pressure regulation

- **Confocal-compatible imaging chamber**: The chamber fits both upright and inverted microscopes.
- **Precise cannula positioning**: Three-axis control and angled holders for low-working distance applications such as optical sectioning.
- **Magnetic perfusion setup**: Quick, easy and adjustable, attachments for fluid control.
- **Temperature Monitor**: With temperature control achieved via superfusion.

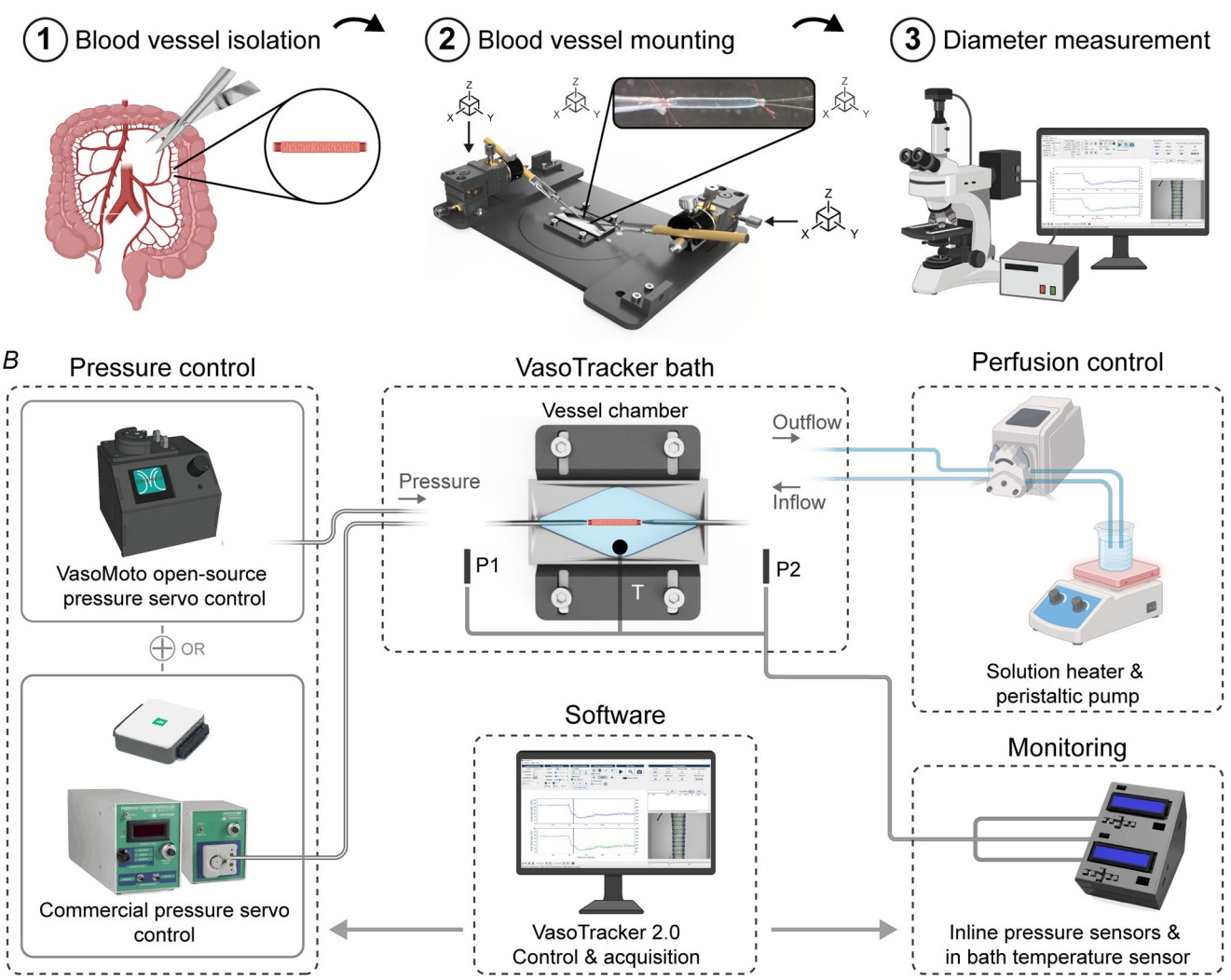

**Figure 4. VasoTracker 2 pressure myography system**
*A*, Schematic of the pressure myography experimental protocol. Blood vessels are freshly isolated from vascular beds of interest (1), then mounted and pressurized in the VasoTracker myograph chamber (2), before being visualized by light microscopy for diameter tracking (3). *B*, SSSchematic of the VasoTracker system. Vessels can be pressurized using the open-source, programmable pressure servo controller, VasoMoto or commercial pressure servo controllers. Bath perfusion, temperature control, and drug delivery are managed through a peristaltic pump and a hot plate or inline heater. Inflow and outflow pressure (P1/P2), as well as temperature (T), are monitored via the Arduino-based VasoTracker 2 pressure/temperature monitor. Image acquisition, diameter measurement and data logging are performed using the VasoTracker software. Components of this figure were created using bioRender.

## Automated pressure response curves (myogenic tone experiments)

Measurement of pressure-response curves is the most common method used to investigate the intrinsic ability of blood vessels to constrict in response to an increase in pressure – myogenic tone (Wilson et al., 2022). Typically, investigators assess the ability of resistance arteries or arterioles to constrict as intraluminal pressure is increased in a stepwise fashion. The experiment is performed under both active and passive conditions to determine the degree of myogenic response at each pressure. The active conditions include calcium in the bath solution, allowing smooth muscle cells to contract, while the passive condition removes calcium to eliminate the ability of smooth muscle cells to constrict.

These experiments are labour-intensive, requiring manual adjustment of pressure and manual recording of pressure changes. To address this, we developed Vaso-Moto, an open-source pressure servo system and peristaltic pump specifically designed for pressure-response curve experiments. When paired with VasoTracker 2 software, VasoMoto allows complete automation of these experiments (Fig. 5*A* and Supplementary Movie 3). The system can adjust intraluminal pressure stepwise at pre-determined intervals, with time and pressure changes automatically recorded in the VasoTracker GUI. Beyond its cost-effectiveness compared to commercial alternatives, VasoMoto offers scalability through its modular design, allowing researchers to adapt the system for different experimental configurations (for example to deliver pulsatile pressure). Importantly, VasoTracker 2 is also compatible with commercial pressure servo systems, offering researchers the flexibility to integrate the software with their existing hardware setups.

Figure 5*B* and *C* shows example data collected using VasoTracker automation. Myogenic responses in mouse (Fig. 5*B*) and rat (Fig. 5*C*) mesenteric arteries were measured using VasoTracker myographs, with pressure protocols that stepped from 20 to 100 (mouse) or to 200 mmHg (rat). Arteries from each species exhibited, and maintained, myogenic constriction at pressures >60 mmHg. Such automation reduces the need for manual intervention, allows for more efficient data collection and increases robustness in the experiment, as the interval between each pressure increase is consistently maintained.

## Confocal fluorescence imaging in pressurized arteries

Confocal imaging is a powerful technique for studying the structure and function of blood vessels under near physiological conditions. High-resolution optically sectioned imaging can reveal detailed structural features of the vessel wall, such as the alignment of vascular cells and the expression pattern of key proteins, which are critical for understanding vascular function (Dora et al., 2021). Optically sectioned imaging can also be used to study intracellular calcium dynamics that drive smooth muscle contraction (Taylor et al., 2023; Worton et al., 2021) and calcium signalling and nitric oxide production in endothelial cells that promote vasodilatation (Garland et al., 2017; Wallis et al., 2023).

VasoTracker 2 enables these advanced imaging capabilities by being fully compatible with both upright and inverted confocal microscopes (Fig. 6*A*–*C*). When used on an upright confocal microscope, the angled cannulas provide the required clearance necessary to accommodate high numerical aperture water-dipping objectives. If used on an inverted confocal microscope, the manipulators allow the vessel to be positioned close to the cover slip on the imaging chamber base so that low working distance oil immersion objectives may be used. These features allow researchers to maintain the artery under pressurized conditions while simultaneously capturing high-resolution optical sections or confocal recordings of calcium activity within smooth muscle cells (Fig. 6*D*) or endothelial cells (Fig. 6*E*) within the vessel wall. An additional feature of the VasoTracker 2 bath is that the modular chamber is designed so that it can replace the stage ring of commercial microscopes, making it suitable for all live cell imaging studies (Fig. 6*F* and *G*).

## Discussion

Tracking blood vessel diameter is fundamental to understanding vascular function in health and disease. Changes in vessel diameter regulate blood flow, vascular resistance and tissue perfusion, whereas disruptions in these mechanisms contribute to conditions such as hypertension, diabetes, neurodegenerative diseases and sepsis. Although researchers employ various methodologies to study vascular dynamics, from isolated vessel preparations to in vivo imaging, all require accurate diameter measurement to generate meaningful insights.

VasoTracker 2 addresses this core need through versatile blood vessel diameter tracking software that works with data from multiple imaging modalities. By supporting both real-time tracking and offline analysis of brightfield, fluorescence and ultrasound images, the platform facilitates diverse applications from pressure myography to *in vivo* imaging studies. We have demonstrated how VasoTracker 2 can be used to assess vascular tone in fluorescence labelled arterial strips (Wilson et al., 2020) and in conduit arteries visualized using ultrasound imaging. However, the tracking algorithms could also be applied to other functional imaging datasets beyond traditional approaches, such as *in vivo* two-photon fluorescence imaging (Longden et al.,

2017), while maintaining a standardized analysis across the field.

The open-source nature of VasoTracker 2 software eliminates the need for expensive proprietary acquisition and analysis tools, democratizing access to sophisticated vascular measurement techniques. Because the software is built on µManager 2 (Stuurman et al., 2010), a popular open source microscopy platform, it integrates

with a wide range of cameras and microscope setups, allowing researchers to enhance existing systems without significant additional investment. This accessibility is particularly valuable in resource-limited settings and for educational institutions training the next generation of vascular physiologists.

For researchers conducting *ex vivo* blood vessel experiments, VasoTracker 2 offers additional benefits

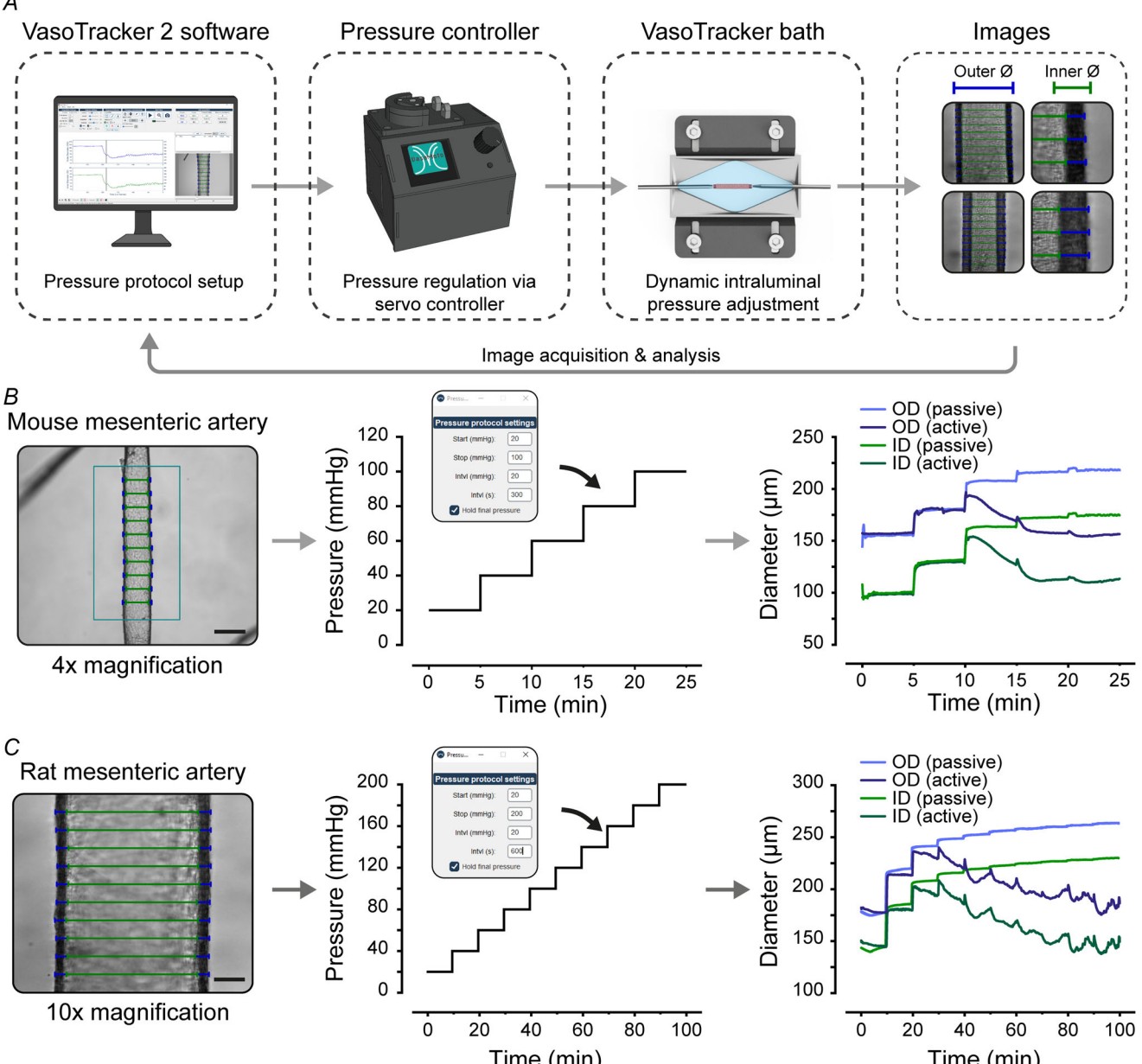

**Figure 5. Automated myogenic tone experiments**
*A*, Schematic of the VasoTracker 2 set up for automated control of intraluminal pressure using the open-source VasoMoto or commercial Living Systems PS-200 pressure servo systems. *B* and *C*, Example images with diameter indicators overlaid (outer diameter; blue, inner diameter; green), automatic pressure step protocols, with Vaso-Tracker 2 pressure settings interface and diameter traces from pressure–curve experiments performed in mouse (*B*) or rat (*C*) mesenteric arteries. Scale bars = 200 µm (*B*) or 50 µm (*C*). Components of this figure were created using bioRender.

through its complementary hardware components. The system provides a fully open-source, low-cost alternative to commercial pressure myography setups. Researchers can construct our second-generation pressure myography system at a cost that is significantly reduced when compared to commercial alternatives. Beyond cost considerations, VasoTracker 2 offers flexibility, allowing users to tailor the system to their specific experimental

needs. The redesigned vessel chamber facilitates advanced imaging techniques including confocal microscopy, whereas VasoMoto, a new open-source peristaltic pump and pressure servo controller, enables automated intraluminal pressure regulation, which may improve experimental reproducibility. Importantly, the modular design allows all VasoTracker 2 components to be used independently, providing researchers with tools for vessel

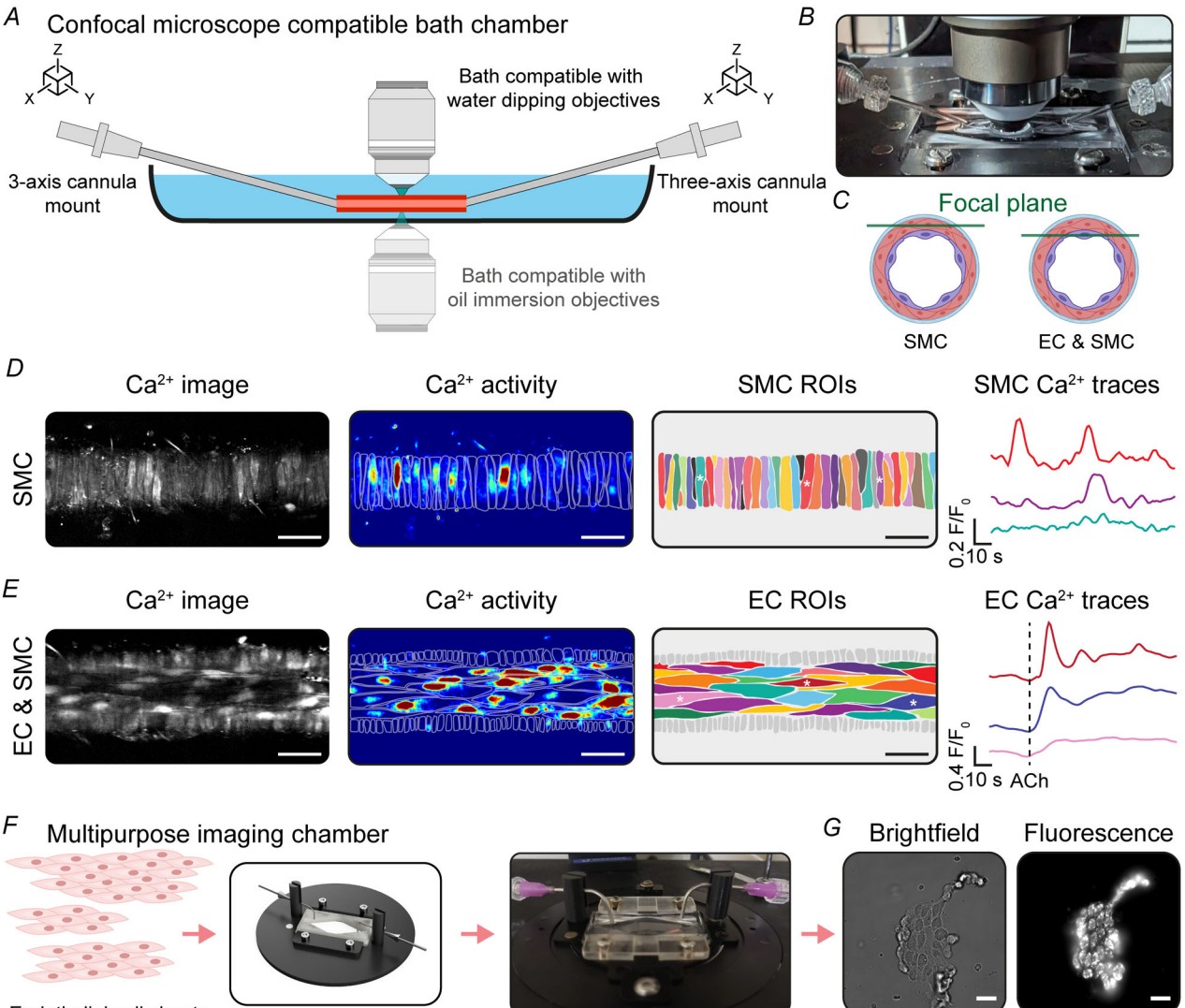

**Figure 6.  Fluorescence calcium imaging in pressurized arteries using confocal imaging**
*A*, The three-axis cannula mounts on VasoTracker chambers facilitate clearance for water dipping objectives on upright microscopes. Additionally, three-axis cannula positioning allows vessels to be located close to the cover slip bottom of the chamber for compatibility with oil-immersion lenses on inverted confocal microscopes. *B*, Image showing the experimental setup on an upright confocal microscope. *C*, Diagram illustrating how optical sectioning facilitates the imaging of the smooth muscle cell (SMC) or endothelial cell (EC) layer in pressurized arteries. *D* and *E*, Representative calcium images and corresponding basal calcium signals from smooth muscle cells (*D*) and acetylcholine-evoked (ACh) (1 µM) activity in endothelial cells (*E*) in rat mesenteric arteries pressurized to 60 mmHg. *F* and *G*, Schematic (*F*) and example fluorescence images (*G*) obtained using the VasoTracker chamber for live cell imaging of acutely isolated endothelial cell sheets. The chamber is designed to replace the microscope stage ring in most commercial microscopes. Cells were isolated from rat mesenteric arteries and loaded with the mitochondrial membrane potential sensitive indicator, Mitotracker red (100 nM). Scale bars = 50 µm (*D* and *E*) or 20 µm (*G*).

imaging and analysis that can be adapted to a wide range of experimental applications beyond pressure myography.

The technical complexity of vascular research techniques represents another barrier that VasoTracker 2 software and hardware addresses through design improvements focused on usability and functionality. The hardware components are designed with simplicity in mind. The majority of components are off-the-shelf-parts and custom components can be purchased from commercial machining companies. If researchers can follow basic assembly instructions, they can construct a VasoTracker 2 system within 1 day. In our teaching laboratories, undergraduate students with brief training have successfully used the software for the analysis of vascular pharmacology experiments, whereas, in our research laboratory, the complete pressure myograph system has been used by students for undergraduate research projects. Recent advancements in guidelines for assessing blood vessel function have provided standardized procedures that, combined with accessible tools like VasoTracker 2, may help new users apply these techniques effectively (Wenceslau et al., 2021).

Importantly, VasoTracker's core innovation is a broadly applicable set of tools to measure blood vessel diameter changes in response to experimental intervention. Although our validation experiments focus on myogenic and pharmacological responses to demonstrate the core tracking functionality, the modular design enables integration with diverse experimental approaches beyond these methods. For example, researchers could readily use our software and/or hardware with one of many open-source electrical stimulators available (Cermak et al., 2019; Ott & Jung, 2023; Sanders & Kepecs, 2014; Sheinin et al., 2015) to study neural vascular control mechanisms. Additionally, laboratories seeking to absolutely minimize costs could pair VasoTracker software or hardware with an open-source microscope solution (Sharkey et al., 2016).

VasoTracker remains an open-source project, which facilitates continued development and community-driven improvements. By making both software and hardware designs freely available, the research community can contribute to its ongoing development, including validation and sharing of protocols for applications such as neural stimulation integration. The adoption of VasoTracker by multiple laboratories indicates the value of accessible research tools, with recent examples including its integration into diagnostic technology such as HemoLens, which incorporates VasoTracker software for diameter tracking in a specialized 3D-printed pressure myography system (PereiraTavares et al., 2025).

Although VasoTracker 2 addresses many challenges in vascular research, limitations remain. We routinely use VasoTracker to track diameter in vessels ranging from 10 µm to 1 mm in diameter. For smaller vessels, we specifically designed the VasoTracker 2 chamber to be compatible with high magnification, high numerical aperture oil immersion objectives. The manipulators allow vessels to be positioned at the bottom of the chamber within the working distance of these objectives. This permits visualization of very small arteries, but automated tracking accuracy is ultimately limited by the optical diffraction limit of the imaging system.

We have demonstrated that the software works well for most vessel types, although extremely small vessels (<10 µm) may present tracking challenges that require manual intervention. VasoTracker software includes manual calliper tools to enable diameter measurement when automated tracking becomes challenging. As with any image analysis software, performance depends on image quality: brightfield microscopy benefits from good contrast between vessel walls and background, fluorescence imaging requires adequate signal-to-noise ratios and ultrasound analysis performance depends entirely on the resolution and quality of the ultrasound system and its video output. These limitations are not unique to VasoTracker, and are the same as those faced by commercial systems and software.

Although VasoTracker 2 is designed for straightforward implementation, some research groups may be hesitant to build their own system rather than purchasing a ready-made commercial alternative. The hardware components are straightforward to assemble with minimal technical skills required. The Arduino-based pressure sensor and VasoMoto control components require a small amount of basic soldering. To support this, full construction guides are available on the VasoTracker website. The software is designed to be user-friendly, with comprehensive user guides, tutorial videos and detailed documentation to support users.

In summary, VasoTracker 2 represents a community-driven approach to vascular research that democratizes access to sophisticated analysis techniques. By providing versatile software for quantitative vascular analysis across multiple imaging modalities, complemented by modular, low-cost hardware components, the platform reduces barriers to entry for vascular research and has the potential to accelerate scientific discovery and foster reproducibility across laboratories worldwide.

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

## Additional information

### Data availability statement

Data supporting the findings of this study are available for download with the VasoTracker software, or from the corresponding authors on request. All code and design files are available via the VasoTracker website and associated repositories (https://www.vasotracker.com).

### Competing interests

The authors declare that they have no competing interests.

### Author contributions

C.W., J.G.M. and M.L. developed the concept. C.W., M.D.L and C.O. wrote the VasoTracker software, whereas C.W. and N.T. developed the Arduino code. C.W. and M.D.L. designed the experimental apparatus. C.W., M.L., A.M., J.A., D.J.M, R.S., G.E., F.B. and D.A.J. performed the experiments. D.A.J, N.T., F.D. and O.H. were alpha testers and contributed valuable feedback. C.W. drafted the manuscript. All authors revised, edited and approved the final version of the manuscript submitted for publication. All authors agree to be accountable for all aspects of the work. All persons designated as authors qualify for authorship, and all those who qualify for authorship are listed.

### Funding

This work was funded by the British Heart Foundation (RG/F/20/110007; PG/20/9/34859), Wellcome (204682/Z/16/Z; 202924/Z/16/Z) and funding from The Strathclyde Institute of Pharmacy and Biomedical Sciences whose support is gratefully acknowledged. Support for work on the project by CO was provided by the University of Strathclyde's Faculty of Science Research Software Engineering group. FD is supported by two research grants from the Ludeman Family Center for Women's Health Research located at the University of Colorado Anschutz Medical Campus (2019 and 2024); two research grants from the University of Pennsylvania Orphan Disease Center in partnership with the cureCADASIL (2019 and 2022), a research grant from the Leducq Foundation for Cardiovascular Research (Leducq Transatlantic Network of Excellence 22CVD01 BRENDA), the National Institute of Neurological Disorders and Stroke (NINDS; RF1NS129022 and RF1NS140137) and the National Heart, Lung and Blood Institute (NHLBI; R01HL136636). DAJ is supported by the National Heart, Lung and Blood Institute (5T32GM007635 and F31HL170645). OFH is supported by the National Heart, Lung and Blood Institute (NHLBI; R01HL169681), the National Institute on Aging (NIA; R21AG082193), the National Institute of General Medical Sciences (NIGMS; P20GM135007), the American Heart Association (20CDA35310097). the Bloomfield Early Career Professorship in Cardiovascular Research, the Totman Medical Research Trust and the Cardiovascular Research Institute of Vermont. NRT is supported by the National Heart, Lung and Blood Institute (P01HL152951) and the National Institute of Diabetes and Digestive and Kidney Diseases (R01DK119615 and R01DK135696). DRM is supported by the National Center for Complementary and Integrative Health (NCCIH; R00 AT010017). We also thank the VasoTracker community for their ongoing contributions to the project.

### Acknowledgements

We thank the VasoTracker community for their valuable feedback, bug reports and contributions to the ongoing development of this open-source platform.

### Keywords

blood vessel diameter, endothelium, smooth muscle, vasodilatation, vasoconstriction, open source

## Supporting information

Additional supporting information can be found online in the Supporting Information section at the end of the HTML view of the article. Supporting information files available:

**Peer Review History**
**Supplementary Movies**
**Supplementary Movies**
**Supplementary Movies**

