## [Peer Review History · The Journal of Physiology]

VasoTracker 2: Open-source Software and Hardware for Tracking Blood Vessel Diameter and Assessing Vascular Function

Matthew D Lee, Christopher Osborne, Ross Stevenson, Amy MacDonald, Grace Ebner, Danielle Jeffrey, Margaret A MacDonald, Xun Zhang, Charlotte Buckley, Fabrice Dabertrand, Daniel R Machin, Jason S Au, Osama F Harraz, Nathan Roger Tykocki, John G McCarron, and Calum Wilson

DOI: 10.1113/JP289322

Corresponding author(s): Calum Wilson (c.wilson@strath.ac.uk)

The following individual(s) involved in review of this submission have agreed to reveal their identity: R Clinton Webb (Referee #1); Aaron Trask (Referee #2)

Review Timeline:

Submission Date:	23-May-2025
Editorial Decision:	24-Jul-2025
Revision Received:	20-Aug-2025
Editorial Decision:	08-Sep-2025
Revision Received:	11-Sep-2025
Accepted:	16-Sep-2025

Senior Editor: Kim Barrett

Reviewing Editor: T Alexander Quinn

Transaction Report:

Dear Dr Wilson,

Re: JP-TFP-2025-289322 "VasoTracker 2: An Open-source Platform for Quantitative Analysis of Vascular Reactivity and Function" by Matthew D Lee, Christopher Osborne, Ross Stevenson, Amy MacDonald, Grace Ebner, Danielle Jeffrey, Margaret A MacDonald, Xun Zhang, Charlotte Buckley, Fabrice Dabertrand, Daniel R Machin, Jason S Au, Osama F Harraz, Nathan Roger Tykocki, John G McCarron, and Calum Wilson

Thank you for submitting your manuscript to The Journal of Physiology. It has been assessed by a Reviewing Editor and by 2 expert referees and we are pleased to tell you that it is potentially acceptable for publication following satisfactory major revision.

LANGUAGE EDITING AND SUPPORT FOR PUBLICATION: If you would like help with English language editing, or other article preparation support, Wiley Editing Services offers expert help, including English Language Editing, as well as translation, manuscript formatting, and figure formatting at www.wileyauthors.com/eoo/preparation. You can also find resources for Preparing Your Article for general guidance about writing and preparing your manuscript at www.wileyauthors.com/eoo/prepresources.

REVISION CHECKLIST:

We look forward to receiving your revised submission.

Yours sincerely,

Kim Barrett
Senior Editor
The Journal of Physiology

REQUIRED ITEMS

- Author photo and profile. First or joint first authors are asked to provide a short biography (no more than 100 words for one author or 150 words in total for joint first authors) and a portrait photograph. These should be uploaded and clearly labelled together in a Word document with the revised version of the manuscript. See Information for Authors for further details.

- You must start the Methods section with a paragraph headed Ethical approval (https://jp.msubmit.net/cgi-bin/main.plex?form_type=display_requirements#methods).

Research must comply with The Journal's policies regarding animal experiments (<https://physoc.onlinelibrary.wiley.com/hub/animal-experiments>) and adherence to these policies must be stated in the manuscript.

Authors should confirm in their Methods section that their experiments were carried out according to the guidelines laid down by their institution's animal welfare committee, including an ethics approval reference number. The Methods section must contain a statement about access to food, water and housing, details of the anaesthetic regime: anaesthetic used, dose and route of administration, and method of killing the experimental animals.

- The reference list must be in alphabetical order, rather than numbered, to comply with our Journal format.

- Your manuscript must include a complete Additional Information section, including competing interests; funding; author contributions and acknowledgements.

- Please upload separate high-quality figure files via the submission form.

- Please ensure that any tables are editable and in Word format, and wherever possible, embedded in the article file itself.

- Please ensure that the Article File you upload is a Word file.

- Please include an Abstract Figure file, as well as the Figure Legend text within the main article file. The Abstract Figure is a piece of artwork designed to give readers an immediate understanding of the research and should summarise the main conclusions. If possible, the image should be easily 'readable' from left to right or top to bottom. It should show the physiological relevance of the manuscript so readers can assess the importance and content of its findings. Abstract Figures should not merely recapitulate other figures in the manuscript. Please try to keep the diagram as simple as possible and without superfluous information that may distract from the main conclusion(s). Abstract Figures must be provided by authors no later than the revised manuscript stage and should be uploaded as a separate file during online submission labelled as File Type 'Abstract Figure'. Please also ensure that you include the figure legend in the main article file. All Abstract Figures should be created using BioRender. Authors should use The Journal's premium BioRender account to export high-resolution images. Details on how to use and access the premium account are included as part of this email.

EDITOR COMMENTS

Reviewing Editor:

This is a well written description of VasoTracker 2, a technique for the measurement of vascular diameters under several experimental conditions.

Overall, the reviewers were positive about the paper, however have made important suggestions for its improvement.

Specifically, the paper needs to describe limitations of the system and how it will impact its utility and influence, as well as its application across imaging modalities with varying specifications and its practical implementation (e.g., related to the required hardware) by other groups.

REFEREE COMMENTS

Referee #1:

This manuscript provides a useful description of the VasoTracker 2 for the measurement of vascular diameters under several experimental conditions.

What's missing:

1. The authors made no effort to describe limitations of the system (physical, technical skills, hardware assembly, etc.) .
2. While myogenic tone and pharmacological approaches are important, I would argue that sympathetic and non-adrenergic, non-cholinergic neural control is just as important. This is a major weakness of the VasoTracker 2 system.

Referee #2:

This is a well-written manuscript describing the development of an updated version of VasoTracker, dubbed VasoTracker 2. The authors described impressive upgrades to the original platform based on user feedback. I only have two minor comments for the authors' consideration. First, how does VasoTracker 2 perform in terms of resolution given that different imaging modalities can have different resolutions (pixel resolutions)? Does it have a universal calibration capability to account for this? Some discussion related to this point would be helpful. Secondly, some commentary about the availability of the hardware would be useful. How would users procure the hardware described in the manuscript if they wanted to?

END OF COMMENTS

REQUIRED ITEMS

- Author photo and profile. First or joint first authors are asked to provide a short biography (no more than 100 words for one author or 150 words in total for joint first authors) and a portrait photograph. These should be uploaded and clearly labelled together in a Word document with the revised version of the manuscript. See Information for Authors for further details.

Response: A short biography and photograph of the first author has been provided.

- You must start the Methods section with a paragraph headed Ethical approval (https://jp.msubmit.net/cgi-bin/main.plex?form_type=display_requirements#methods).

- Research must comply with The Journal's policies regarding animal experiments (<https://physoc.onlinelibrary.wiley.com/hub/animal-experiments>) and adherence to these policies must be stated in the manuscript.

- Authors should confirm in their Methods section that their experiments were carried out according to the guidelines laid down by their institution's animal welfare committee, including an ethics approval reference number. The Methods section must contain a statement about access to food, water and housing, details of the anaesthetic regime: anaesthetic used, dose and route of administration, and method of killing the experimental animals

Response: Our Methods section now begins with the required "Ethical approval" paragraph detailing institutional approvals and compliance with The Journal's animal experiment policies, UK Home Office regulations, and ARRIVE guidelines 2.0. The Animals section (under Experimental Protocols) includes the required details about housing, anaesthetic regimes (doses and routes), and euthanasia methods as specified in the guidelines.

- - The reference list must be in alphabetical order, rather than numbered, to comply with our Journal format.

Response: Done.

- Your manuscript must include a complete Additional Information section, including competing interests; funding; author contributions and acknowledgements.

Response: Done.

- Please upload separate high-quality figure files via the submission form.

Response: Done.

- Please ensure that the Article File you upload is a Word file.

Response: Done.

- Please include an Abstract Figure file, as well as the Figure Legend text within the main article file. T

Response: Done.

Reviewing Editor:

This is a well written description of VasoTracker 2, a technique for the measurement of vascular diameters under several experimental conditions.

Overall, the reviewers were positive about the paper, however have made important suggestions for its improvement.

Specifically, the paper needs to describe limitations of the system and how it will impact its utility and influence, as well as its application across imaging modalities with varying specifications and its practical implementation (e.g., related to the required hardware) by other groups.

Response: We thank the editor and reviewers for their positive feedback on our description of VasoTracker 2. We have carefully addressed the specific concern regarding limitations and practical implementation considerations.

We have expanded the methods to include additional details and added discussion of additional practical considerations that provides a comprehensive overview of:

- System parameters and performance boundaries: In the methods, we now clearly specify the resolution and field of view achievable with the camera and microscope systems we use, whilst noting that these are not limitations of VasoTracker per se, but microscopy in general. Furthermore, in the discussion we highlight challenges with extremely small vessels (<10 μm), and note that VasoTracker provides manual caliper tools for when automated tracking becomes challenging.
- Cross-modal imaging considerations: We have detailed how performance varies across brightfield microscopy, fluorescence imaging, and ultrasound analysis, clarifying that image quality requirements depend on the inherent characteristics of each modality rather than limitations of VasoTracker 2 itself. For ultrasound specifically, we emphasize that analysis performance depends entirely on the resolution and quality of the ultrasound system and its video output.
- Practical implementation considerations: We address the potential hesitation some research groups may have about building their own system rather than purchasing commercial alternatives. We clarify that hardware assembly requires only basic skills and minimal soldering. We also emphasize that comprehensive user guides, tutorial videos, and detailed documentation are provided to support implementation.

These additions provide readers with realistic expectations about VasoTracker 2's capabilities and requirements, helping them assess its suitability for their specific research needs while highlighting the extensive support materials provided to facilitate successful adoption.

We also have included a potential cover art image featuring segmented cells from confocal imaging experiments using VasoTracker 2. We believe this would make an excellent journal cover and hope you will consider featuring it.

Referee #1:

This manuscript provides a useful description of the VasoTracker 2 for the measurement of vascular diameters under several experimental conditions.

Response: We thank the reviewer for the positive evaluation of our manuscript.

1. The authors made no effort to describe limitations of the system (physical, technical skills, hardware assembly, etc.).

Response: In our initial manuscript, we provided a paragraph in the discussion acknowledging the main limitations of the VasoTracker (tracking small vessels and technical skills for hardware assembly):

“While VasoTracker 2 addresses many challenges in vascular research, limitations remain. While the software works well for most vessel types, extremely small vessels (< 10 μm) or vessels with low contrast may present tracking challenges. Additionally, the hardware components currently require some, albeit minimal, technical assembly skills, and future developments could focus on further simplifying construction. These limitations provide opportunities for future improvements as the open-source community continues to develop the platform.”

We have now expanded this section to two paragraphs to provide additional discussion of issues such as image resolution (usually limited by the diffraction limit), usability, and technical assembly requirements. We also highlight how we have attempted to minimise barriers to entry by providing extensive documentation, instructions, and even tutorial videos (attached as supplementary information for review, but also available on the VasoTracker website). In addition, throughout the methods we have added additional technical details that will help potential users assess VasoTracker’s suitability for their specific research needs.

2. While myogenic tone and pharmacological approaches are important, I would argue that sympathetic and non-adrenergic, non-cholinergic neural control is just as important. This is a major weakness of the VasoTracker 2 system.

Response: We thank the reviewer for highlighting the importance of sympathetic and NANC neural control in vascular physiology. We also appreciate the opportunity to clarify that VasoTracker 2 is designed as a broadly applicable platform. The absence of neural stimulation examples in our validation does not represent a limitation of VasoTracker 2, but reflects our own laboratories' research focus on myogenic and pharmacological control. Our validation experiments focusing on these approaches effectively demonstrate the software's core tracking algorithms across multiple imaging modalities.

VasoTracker 2's core innovation is a versatile set of tools (software and optional hardware) that can measure vascular diameter changes in response to any experimental intervention, including neural mechanisms. While we do not design a dedicated "VasoTracker Stimulator," we have highlighted in our discussion that researchers can integrate VasoTracker 2 with existing electrical stimulation systems. Inexpensive open-source stimulators like StimJim and MyoPulser are excellent examples that the extended research community could integrate into VasoTracker 2. Our open-source software approach also allows experimenters to fully integrate software control for such systems, enabling the comprehensive neural vascular control studies.

Referee #2:

This is a well-written manuscript describing the development of an updated version of VasoTracker, dubbed VasoTracker 2. The authors described impressive upgrades to the original platform based on user feedback. I only have two minor comments for the authors' consideration.

Response: We thank the reviewer for the positive comments.

First, how does VasoTracker 2 perform in terms of resolution given that different imaging modalities can have different resolutions (pixel resolutions)? Does it have a universal calibration capability to account for this? Some discussion related to this point would be helpful.

Response: We thank the reviewer for this important question. VasoTracker 2 includes calibration capabilities to handle varying pixel resolutions across different imaging systems. Users calibrate the system using reference objects of known dimensions (e.g., calibration slides, microbeads, or cannulation needles of known diameter) to establish the pixel-to-micron conversion factor for their specific imaging setup.

This calibration approach works universally across brightfield microscopy, fluorescence imaging, and ultrasound systems, regardless of the native pixel resolution of each modality. Once calibrated, users can save these settings for different imaging configurations and quickly switch between them.

We have expanded the methods section to include additional details about the calibration procedure (see *Calibration*). Additionally, we have included additional comments in the *Choice of Imaging System* for the myography section ensuring users understand how to achieve accurate measurements regardless of their specific hardware configuration.

Secondly, some commentary about the availability of the hardware would be useful. How would users procure the hardware described in the manuscript if they wanted to?

Response: We thank the reviewer for this important point about hardware procurement. We have addressed this point by adding a new "System Overview and Availability" section to the Methods that clearly explains how users can access the hardware designs and components. As detailed in this new section, all hardware design files, comprehensive component lists, and assembly instructions are freely available through the VasoTracker website and GitHub repository. Users can source the required components from standard suppliers (electronics distributors, machining services, etc.) and either fabricate parts themselves or use commercial machining services. The modular design allows researchers to build only the components they need or integrate individual elements with existing laboratory equipment.

Dear Dr Wilson,

Re: JP-TFP-2025-289322R1 "VasoTracker 2: Open-source Software and Hardware for Tracking Blood Vessel Diameter and Assessing Vascular Function" by Matthew D Lee, Christopher Osborne, Ross Stevenson, Amy MacDonald, Grace Ebner, Danielle Jeffrey, Margaret A MacDonald, Xun Zhang, Charlotte Buckley, Fabrice Dabertrand, Daniel R Machin, Jason S Au, Osama F Harraz, Nathan Roger Tykocki, John G McCarron, and Calum Wilson

Thank you for submitting your manuscript to The Journal of Physiology. It has been assessed by a Reviewing Editor and by 2 expert referees and we are pleased to tell you that it is acceptable for publication following satisfactory revision.

REVISION CHECKLIST:

Please upload two versions of your manuscript text: one with all relevant changes highlighted and one clean version with no changes tracked. The manuscript file should include all tables and figure legends, but each figure/graph should be uploaded as separate, high-resolution files. The journal is now integrated with Wiley's Image Checking service. For further details,

see: <https://www.wiley.com/en-us/network/publishing/research-publishing/trending-stories/upholding-image-integrity-wileys-image-screening-service>

We look forward to receiving your revised submission.

Yours sincerely,

Kim Barrett
Senior Editor
The Journal of Physiology

EDITOR COMMENTS

Reviewing Editor:

All concerns of the reviewers have been satisfied, however statements of local ethical approval for the human and animal experiments are missing.

Ethics Concerns:

Local ethical approval for both the human and animal experiments has not been stated.

REFEREE COMMENTS

Referee #1:

No further comments.

Referee #2:

The authors have addressed my previous comments.

END OF COMMENTS

Response to Referees

Reviewing Editor:

All concerns of the reviewers have been satisfied, however statements of local ethical approval for the human and animal experiments are missing.

Ethics Concerns: Local ethical approval for both the human and animal experiments has not been stated.

Response: Thank you for this feedback. We have ensured that our ethical approval statement is clearly presented at the start of our "Experimental Protocols" section. Our manuscript includes comprehensive ethical approval information covering all animal work (approved by University of Strathclyde Animal and Welfare Ethical Review Committee, University of Colorado Anschutz Medical Campus IACUC, University of Vermont IACUC, and University of Utah/Salt Lake City Veterans Affairs Medical Center Animal Care and Use Committee) and human work (approved by University of Waterloo ethics board, ORE 22477), all conducted in accordance with relevant national guidelines, the Declaration of Helsinki, and ARRIVE guidelines 2.0.

Dear Dr Wilson,

Re: JP-TFP-2025-289322R2 "VasoTracker 2: Open-source Software and Hardware for Tracking Blood Vessel Diameter and Assessing Vascular Function" by Matthew D Lee, Christopher Osborne, Ross Stevenson, Amy MacDonald, Grace Ebner, Danielle Jeffrey, Margaret A MacDonald, Xun Zhang, Charlotte Buckley, Fabrice Dabertrand, Daniel R Machin, Jason S Au, Osama F Harraz, Nathan Roger Tykocki, John G McCarron, and Calum Wilson

We are pleased to tell you that your paper has been accepted for publication in The Journal of Physiology.

Authors should note that it is too late at this point to offer corrections prior to proofing. Major corrections at proof stage, such as changes to figures, will be referred to the Editors for approval before they can be incorporated. Only minor changes, such as to style and consistency, should be made at proof stage. Changes that need to be made after proof stage will usually require a formal correction notice.

All queries at proof stage should be sent to: TJP@wiley.com

If you would like to receive our 'Research Roundup', a monthly newsletter highlighting the cutting-edge research published in The Physiological Society's family of journals (The Journal of Physiology, Experimental Physiology and Physiological Reports), please click this link, fill in your name and email address and select 'Research Roundup':
<https://www.physoc.org/journals-and-media/membernews/>

Yours sincerely,

Kim Barrett
Senior Editor
The Journal of Physiology

P.S. - You can help your research get the attention it deserves! Check out Wiley's free Promotion Guide for best-practice recommendations for promoting your work at www.wileyauthors.com/eoo/guide. You can learn more about Wiley Editing Services which offers professional video, design, and writing services to create shareable video abstracts, infographics, conference posters, lay summaries, and research news stories for your research at www.wileyauthors.com/eoo/promotion.

IMPORTANT NOTICE ABOUT OPEN ACCESS: To assist authors whose funding agencies mandate public access to published research findings sooner than 12 months after publication The Journal of Physiology allows authors to pay an Open Access (OA) fee to have their papers made freely available immediately on publication.

You can check if your funder or institution has a Wiley Open Access Account here: <https://authorservices.wiley.com/author-resources/Journal-Authors/licensing-and-open-access/open-access/author-compliance-tool.html>

EDITOR COMMENTS

Reviewing Editor:

Thank you for addressing all editorial concerns.